# Autonomic Nervous System Influences on Cardiovascular Self-Organized Criticality

**DOI:** 10.3390/e25060880

**Published:** 2023-05-30

**Authors:** Jacques-Olivier Fortrat, Guillaume Ravé

**Affiliations:** 1CHU Angers, Médecine Vasculaire, INSERM, CNRS, MITOVASC, Equipe CarMe, SFR ICAT, Université d’Angers, 49933 Angers, France; 2Toulouse Football Club, 1 Allée Gabriel Biénès, 31400 Toulouse, France; rave.guillaumme@gmail.com

**Keywords:** autonomic nervous system, baroreflex, blood pressure regulation, cardiovascular dynamics, heart rate variability, homeostasis, fractal, self-organized criticality, soccer, Zipf’s law

## Abstract

Cardiovascular self-organized criticality has recently been demonstrated. We studied a model of autonomic nervous system changes to better characterize heart rate variability self-organized criticality. The model included short and long-term autonomic changes associated with body position and physical training, respectively. Twelve professional soccer players took part in a 5-week training session divided into “Warm-up”, “Intensive”, and “Tapering” periods. A stand test was carried out at the beginning and end of each period. Heart rate variability was recorded beat by beat (Polar Team 2). Bradycardias, defined as successive heart rates with a decreasing value, were counted according to their length in number of heartbeat intervals. We checked whether bradycardias were distributed according to Zipf’s law, a feature of self-organized criticality. Zipf’s law draws a straight line when the rank of occurrence is plotted against the frequency of occurrence in a log–log graph. Bradycardias were distributed according to Zipf’s law, regardless of body position or training. Bradycardias were much longer in the standing position than the supine position and Zipf’s law was broken after a delay of four heartbeat intervals. Zipf’s law could also be broken in some subjects with curved long bradycardia distributions by training. Zipf’s law confirms the self-organized nature of heart rate variability and is strongly linked to autonomic standing adjustment. However, Zipf’s law could be broken, the significance of which remains unclear.

## 1. Introduction

Maintaining internal conditions is a primary physiological challenge for all organisms, and this is achieved through the stability of body constants [1,2]. It is commonly believed that this stability is achieved by sensors involved in regulatory loops with negative feedback, which is known as the homeostasis principle [1,2]. However, the homeostasis principle fails to explain how an organism adjusts to various demands posed by environmental conditions [1,3,4,5]. Physiological system adjustability requires cybernetic alterations of each regulatory loop, referred to as resetting, which is complex to integrate into the homeostasis principle [6].

An alternative concept for physiological homeostasis has been proposed from the field of physics. In the late 1980s, Per Bak identified a pile of sand as a dynamic system spontaneously poised at equilibrium without the need to tune any control parameter to a precise value [7,8,9]. The pile of sand was poised at equilibrium but near to criticality, which is referred to as self-organized criticality. This concept was soon applied to many dynamic systems across various fields, including geophysics, astronomy, sociology, and neurobiology [3,4,5,7,8,9,10].

The interesting point is that systems near to criticality can more easily adjust to changes in environmental conditions [4,5,10]. Criticality means that sudden major events, known as catastrophes or avalanches, may spontaneously occur. In the case of the sand pile model, these major events are sand avalanches, whereas in geophysics, they are earthquakes. According to the Gutenberg–Richter law, the distribution of earthquakes by their magnitude and frequency draws a straight line in a log–log graph [7]. Straight lines such as these in log–log graphs or power laws are features of self-organized criticality [7].

In the cardiovascular system, major events include vasovagal syncope, which may occur spontaneously in the standing position, even in healthy individuals [11]. We previously demonstrated that spontaneous vasovagal events observed on any cardiovascular monitoring follow the Gutenberg–Richter law [12]. We also showed that focusing on heart rate can reveal cardiovascular self-organized criticality, as decreasing heart rate is a part of a vasovagal event [13]. We refer to these decreasing heart rates as “spontaneous bradycardia” and demonstrated that they follow another power law, Zipf’s law [13]. This law states that the longer the bradycardia, the less frequent its occurrence, and that this relationship presents a perfect straight line on a log–log graph.

Cardiovascular adjustment to the standing position is mainly due to the autonomic nervous system [11,14]. To better understand cardiovascular self-organized criticality, we studied Zipf’s law in a model of autonomic nervous system changes. This model includes short-term autonomic changes due to alterations in position, as well as long-term changes due to levels of physical training.

## 2. Materials and Methods

### 2.1. Subjects

The study involved twelve professional male soccer players, with an average age of 24.9 ± 1.5 years (mean ± SEM). The players had normal characteristics, with an average height of 1.77 ± 0.03 m and weight of 75 ± 1.3 kg. Informed consent was obtained from all participants, and the procedures were approved by the Ethics Committee of Angers, France (#2014-36, 30 April 2014) in accordance with the Declaration of Helsinki, Finland.

### 2.2. Experiment

Heart rate variability monitoring was achieved by using a sports watch on a beat-by-beat basis with a one-millisecond resolution for RR interval measurements (the interval between two consecutive heartbeats, Polar Team 2 Heart Rate Monitor). Continuous heart rate recordings were obtained during stand tests that consisted of a ten-minute supine period followed by a seven-minute period in the standing position. The stand tests were performed at 8:30 am, before breakfast. Four stand tests were conducted for each player during a five-week preparation period for the competition season, after the season break. The preparation period included three phases: a two-week “Warm-Up” period, followed by a two-week “Intensive” training period, and finally a one-week “Tapering” period. The first standing test took place after the season break and before the training period. The three subsequent standing tests were conducted at the end of each phase of the training period. Individual performance was not assessed in this experiment due to the absence of a known objective measurement to evaluate it within the context of a so-called team intermittent sport. In soccer, performance relies primarily on abilities rather than aerobic capacity and is influenced not only by physical fitness but also by situational factors such as tactics, scoring, and refereeing [15].

### 2.3. Data Analysis

Heart rate recordings were analyzed during stand tests to obtain the mean heart rate in both the supine and standing positions. Additionally, five-minute heart rate recordings were analyzed in both positions to detect bradycardia sequences and determine their distribution according to Zipf’s law. A bradycardia sequence was defined as successive heartbeat intervals with a decreasing heart rate value. For each five-minute recording, bradycardia sequences were counted and classified by their length in number of heartbeat intervals. The rank of bradycardia sequences of the same length was determined by classifying them according to their frequency of occurrence. A natural log–log diagram was plotted for each five-minute heart rate recording, with the number of sequences on the x-axis and their rank on the y-axis. Linear regression was performed for each diagram to obtain correlation coefficients and slopes. In cases where diagrams showed a tipping point, its position was determined by the best linear fit.

### 2.4. Statistics

Data were presented as the mean ± standard error of the mean (SEM). Statistical analyses were conducted using Prism 8 software (GraphPad Software, San Diego, CA, USA). We considered that a Zipf’s graph distribution would fit a straight line when the absolute value of the correlation coefficient |r| exceeded 0.95, as this high cut-off is commonly used to test for power laws [7,12,13]. The normality of variable distributions was assessed using D’Agostino–Pearson omnibus K2 tests. Pearson correlations were performed when necessary. Linear mixed models were employed to model the relationship between position and training, followed by Tukey’s multiple comparison tests when applicable. Statistical significance was set at *p* < 0.05.

## 3. Results

The quality of the data collected during this field experiment was good, as demonstrated in Figure 1.

Heart rate decreased with each stage of training; the decrease was significant from the end of the warm-up period but remained unchanged thereafter (Figure 2). As expected, standing position increased heart rate, but standing heart rate decreased with training and followed the same pattern as supine heart rate (Figure 2). These changes in heart rate indicate changes in the autonomic nervous system resulting from both training and standing position, as previously reported and as demonstrated by spectral analysis during this experiment [16].

The number of heartbeat intervals involved in bradycardia remained unchanged during the training period (Table 1). However, this number was slightly but significantly lower in the standing position (Table 1).

The bradycardia distribution diagrams showed a short range of distribution in the supine position (Figure 3a), extending to 5.2 ± 0.4 heartbeat intervals. A straight line was observed in nine out of twelve players with |r| > 0.95. The |r| values of the three remaining players who did not conform to a straight line were 0.93, 0.94, and 0.94.

The range of bradycardia distribution was much larger in the standing position, extending to 10.9 ± 0.9 heartbeat intervals (Figure 3b). A tipping point was observed in all twelve players in the standing position, with two lines: one included the first four heartbeat intervals and the other included bradycardia longer than four heartbeat intervals (|r| = 0.98 ± 0.00 and 0.97 ± 0.00, respectively, Figure 3b). Training did not lead to significant changes in the distribution of these longer bradycardias (|r| values were T0: 0.98 ± 0.01, T1: 0.97 ± 0.01, T2: 0.98 ± 0.01, T3: 0.98 ± 0.00). However, the distribution of the long bradycardias was clearly curved in two players in the standing position after the warm-up, with |r| values less than 0.95 (0.90 and 0.94, Figure 3c). The short bradycardias remained distributed according to a straight line in these two cases.

The slope of the short bradycardia distribution was not affected by training, while position had a significant impact (Figure 4).

Given the similarity in the strong effect of position on this slope and that of heart rate, we sought to establish a correlation between these two variables. We observed a high correlation between position and heart rate (Figure 5). The slope of the long bradycardia distribution could only be studied in the standing position due to the limited range of bradycardia distribution in the supine position (refer to Figure 3a). Throughout the experiment, the slope remained consistent (−0.26 ± 0.02; −0.22 ± 0.06; −0.28 ± 0.04; −0.27 ± 0.02 at T0, T1, T2, and T3, respectively). No correlation was found between the slope of the long bradycardia sequence and heart rate (r^2^ = 0.04; *p* = 0.24).

## 4. Discussion

The primary finding of this study is that changes to the autonomic nervous system in response to body position strongly influence cardiovascular self-organized criticality. Specifically, the use of Zipf’s law provides evidence of the impact of standing position on heart rate variability recordings.

Professional soccer players are high-performance athletes and cannot be compared to the general population. However, training high-level sport teams provides an exceptional opportunity to conduct long-term experiments on highly motivated and relatively homogeneous groups of healthy individuals. Monitoring heart rate variability in athletes is important for optimizing training, making it an excellent opportunity for studying long-term cardiovascular adjustments [17]. The present study began after a season break when the players were untrained and deconditioned. However, the subsequent training period significantly improved their physical condition, as indicated by the decrease in heart rate. The changes in heart rate also demonstrated the expected autonomic changes resulting from training [18].

The description of heart rate variability in the 1980s challenged the traditional foundation of physiology based on homeostasis, negative feedback regulatory loops, and the stability of bodily constants [18,19]. Homeostasis faced further challenges when Kobayashi and Musha described the fractal patterns of heart rate variability and when further studies revealed multifractality [20,21]. Several alternative views to the principle of homeostasis have been proposed [1,3,4,5,10], among which self-organized criticality is currently the most compelling [3,4,5,10]. Self-organized criticality explains the fractal patterns observed in heart rate variability and how a dynamic system operating at criticality can adjust to external changes more effectively than a system relying solely on regulatory loops with negative feedback [1,3,4,5,10].

There is no single robust way to demonstrate the self-organized criticality of a dynamic system [8,9], as the evidence may be based on several factors. The analysis of transitions between two states, such as the shift from a supine to a standing position, can provide valuable insights, as suggested by Mukli et al. [22]. Unfortunately, the experimental setup required for such analyses was not feasible in the field experiment we are reporting here. However, we have previously demonstrated critical phase transitions during position shifts by replicating, on the cardiovascular system, the seminal study on movement control conducted by Shöner and Kelso [23,24]. Bak et al., who initially described self-organized criticality, focused on the power law observed in earthquakes, known as Gutenberg–Richter’s law, which has also been detected in cardiovascular dynamics [12,25]. Others have concentrated on Zipf’s law, initially described in language dynamics. Several methods have been employed to demonstrate this law in cardiovascular dynamics [13,26,27,28]. Lo et al. demonstrated self-organized criticality in sleep–wake rhythms by analyzing not only the distribution of brief awakening episodes but also the sleep duration between these episodes [29]. A similar approach was employed by Racz et al. to argue that connectivity dynamics in the prefrontal cortex reflect the critical state of a brain function [21]. This kind of method could potentially be applied to heart rate variability time series, but would require significantly longer recordings than the 5-min recordings conducted in our study. Conducting longer recordings in future studies would also provide an opportunity to validate the power law relationship using more advanced techniques, as proposed by Clauset et al., instead of relying solely on linear regression [30].

Most studies examining Zipf’s law in heart rate variability have used Holter recordings [26,27,28]. A Holter device records the electrocardiogram over a period of at least 24 h during daily life activities. However, this type of recording includes variability that is a response to daily life events. In our study, we focused on quiet, stationary subjects to eliminate variability due to daily activities. The remaining variability reflects the intrinsic regulatory mechanisms and delays of the cardiovascular system. Holter recordings also do not distinguish between different body positions. In a previous study, we demonstrated the effect of spontaneous movements, fidgeting, and iterative body position shifts on heart rate variability [31]. Amaral et al. confirmed that heart rate variability complexity remains high in subjects who are motionless [32]. In the present study, we aim to account for body position as the standing position leads to well-defined autonomic nervous system adjustments [14,16,18]. Our previous work has also linked criticality and instability in heart rate variability to the pathophysiology of vasovagal syncope, a health problem strongly related to standing position [12,13].

Our study provides clear evidence that Zipf’s law is observed in heart rate variability for the standing position, while the results were less conclusive for the supine position. The range of straight lines is limited to a delay of a few beats in the former position. Moreover, alterations of the slope of these bradycardia sequences limited to few beats probably only reflect heart rate alterations since these two variables are strongly correlated. On the contrary, the slope of long bradycardia is not correlated with heart rate and this slope could indicates changes in criticality. This point is supported by the findings of our previous research on vasovagal syncope, which showed that patients with syncope have an increased slope of long bradycardia distribution [13].

The effects of autonomic nervous system adjustments on heart rate due to training are typically noticeable, but their impact on heart rate variability is more challenging to observe, regardless of the analysis tool used [16,18]. In our study, the subtle autonomic changes that occur with training were also not detected by the self-organized criticality assessment tool. However, in two players, Zipf’s law was clearly violated after the warm-up period.

We have previously reported that the cardiovascular Zipf’s law is violated at a delay of four heartbeat intervals, with the underlying reasons remaining unclear [13]. Our recent study confirms this law is broken, but suggests that the autonomic nervous system is not involved. It is possible that this broken law is simply a finite side effect.

Our study provides robust evidence for the existence of Zipf’s law in heart rate variability, supporting the notion of cardiovascular self-organized criticality. However, we also observed instances where Zipf’s law was broken, emphasizing the need for further research to elucidate its significance. Moreover, our study introduces a user-friendly tool for characterizing the intricate dynamics of the cardiovascular system, which could potentially aid in the diagnosis of heart rhythm disorders in clinical settings [33].

## Figures and Tables

**Figure 1 entropy-25-00880-f001:**
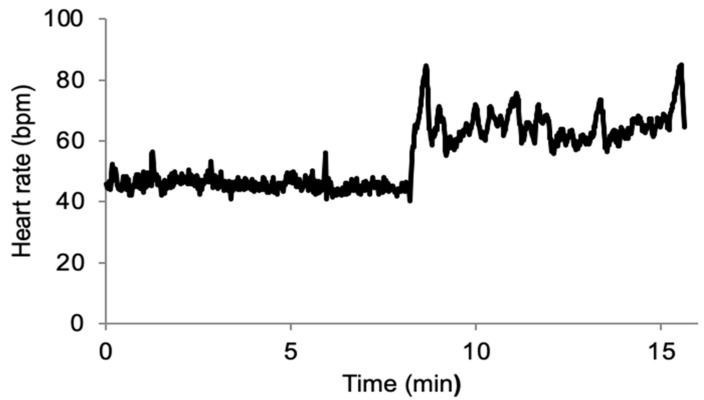
Example of beat-to-beat heart rate recording in a single player during a stand test following ectopic beat removal. The recording starts in the supine position and then transitions to the standing position. bpm: beats per minute.

**Figure 2 entropy-25-00880-f002:**
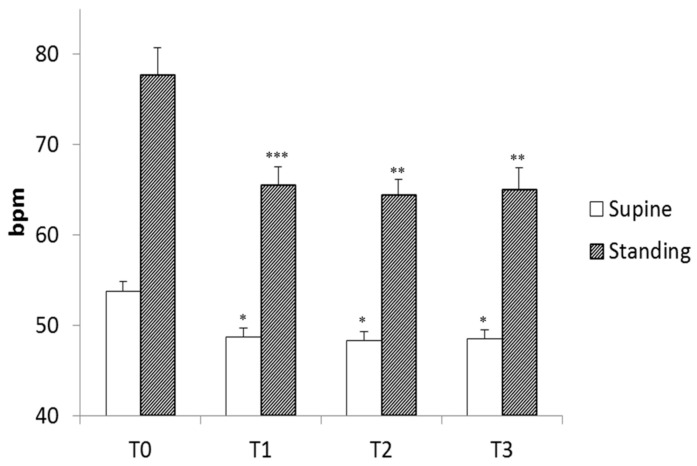
Mean heart rate of the players during four stand tests (T0 to T3) in the supine and standing positions. T0 was the first stand test before the training period; T1 was after the “Warm-Up” period; T2 was after the “Intensive” period; and T3 was after the “Tapering” period. bpm: beats per minute, *: *p* < 0.05; **: *p* < 0.01; ***: *p* < 0.001 vs. T0, linear mixed model followed by Tukey’s multiple comparison test.

**Figure 3 entropy-25-00880-f003:**
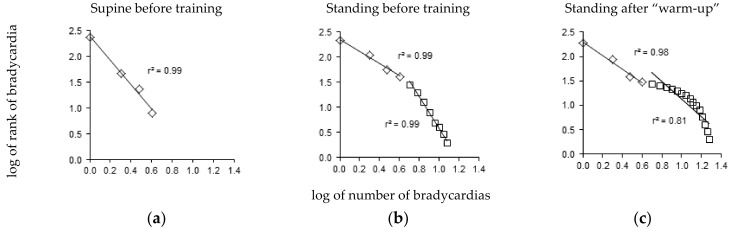
Zipf’s distribution of bradycardia sequences. (**a**) In the supine position before the training period (T0), nine out of twelve players displayed a straight line pattern. (**b**) In the standing position before the training period (T0), all twelve players displayed a pattern of two straight lines with a tipping point at four heartbeat intervals. (**c**) In the standing position after the “Warm-Up” period (T1), only two players displayed a pattern of a curved distribution of bradycardia longer than four heartbeat intervals. Each panel displays a representative example from a single player.

**Figure 4 entropy-25-00880-f004:**
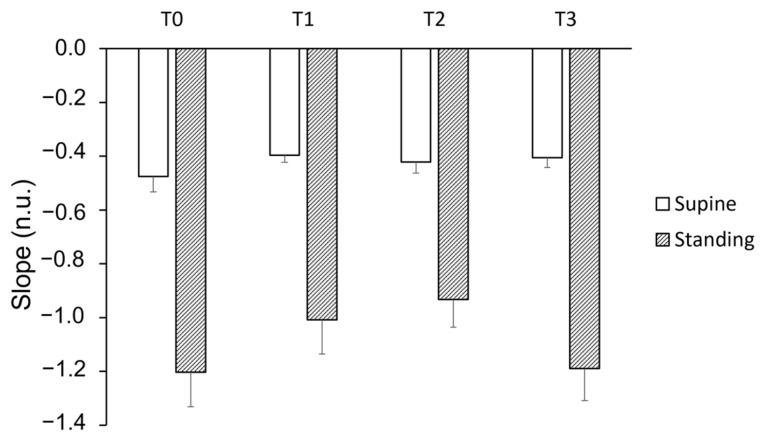
Slope of the distribution of short bradycardia sequences during four stand tests (T0 to T3) in both supine and standing positions. T0 represents the initial stand test conducted before the training period, T1 corresponds to the stand test after the “Warm-Up” period, T2 refers to the stand test following the “Intensive” period, and T3 indicates the stand test conducted after the “Tapering” period. A linear mixed model analysis revealed a lack of effect of training but a strong effect of position (*p* < 0.0001).

**Figure 5 entropy-25-00880-f005:**
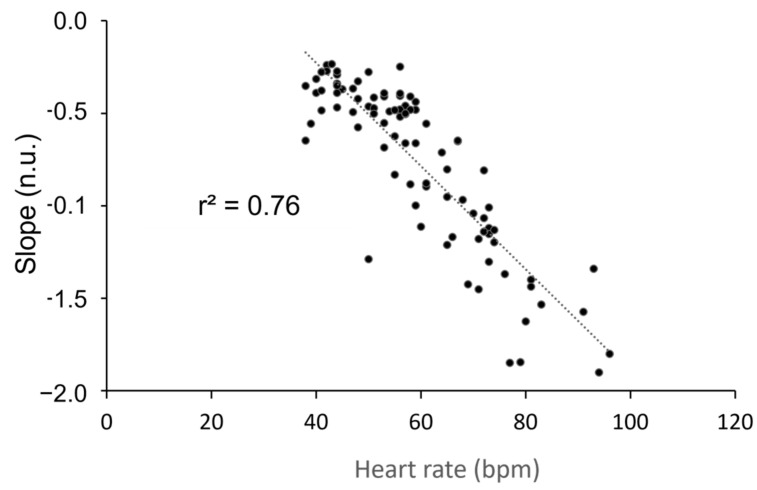
Correlation between heart rate in beats per minute (bpm) and the slope of the distribution of the short bradycardia sequences (no unit, n.u.) during four stand tests performed during a five-week training period. The Pearson’s correlation was significant (*p* < 0.001).

**Table 1 entropy-25-00880-t001:** Normalized number of heartbeat intervals involved in bradycardia sequences during five-minute recordings in supine and standing positions. The number of heartbeat intervals is normalized by the total number of heartbeat intervals in the entire recording. Recordings were conducted at each stage of a five-week preparation for a competition season: before the start of preparation, at the end of the warm-up period, at the end of intensive training, and at the end of the tapering period (T0, T1, T2, and T3, respectively).

	T0	T1	T2	T3
Supine	0.76 ± 0.04	0.71 ± 0.06	0.71 ± 0.06	0.71 ± 0.04
Standing	0.61 ± 0.03	0.63 ± 0.02	0.64 ± 0.03	0.68 ± 0.05

## Data Availability

The data presented in this study are available on reasonable request from the corresponding author.

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
