# Peer review of "Autonomic Nervous System Influences on Cardiovascular Self-Organized Criticality"

_entropy, 2023, doi:10.3390/e25060880_

Round 1
Reviewer 1 Report
Comments on “Autonomic nervous system influences on cardiovascular self-organized criticality”
On the one hand, I consider the study well done and grounded on previous reports. On the other hand, it seems to me that the authors make claims regarding “homeostasis” and “criticality” that are based on a narrow view of homeostasis, and ultimately offer no clear explanation to what is “different” and of what is “gained” in terms of physiological regulation (see comments below). Anyway, these do not compromise the quality of the study and its merit.
Minor points
1. It is common in the medical area to use the term “adaptation” to refer to short-time physiological adjustments, even though, strictly, adaptation refers to the genetic changes in a population on the long-time range.
2. The gender of the subjects is missing in their description.
3. The authors define their bradycardia index as: “A bradycardia sequence was defined as successive heartbeat intervals with a decreasing heart rate value.” However, a few lines later, they state: “This number could be high as a single heartbeat interval may be involved in several bradycardias based on their duration”. Notice that this second phrase is not completely consistent with the definition and the procedure to classify bradycardic events in the study. This sentence should be reviewed.
4. “Further studies are needed to better understand cardiovascular self-organized criticality and its pathophysiological implications. Fortunately, our study also shows that self-organized criticality can be studied easily with a simple sports watch that records heart rate, combined with a straightforward analysis. We identified, counted, and classified the spontaneous bradycardia on heart rate recordings.”. While the first sentence makes reference to the abstract/conceptual understanding of a phenomenon, the second sentence translates this problem as a methodological issue and the last one seems to come out of the blue in this context. This paragraph should be reviewed.
Not so minor points
1. “A tipping point was observed in all twelve players in the standing position, with two lines: one included the first four heartbeat intervals … The short bradycardias remained distributed according to a straight line in these two cases.”
The distribution of bradycardias up to 5 intervals remains as straight lines in the log-log plot in both supine and standing positions, however with different slopes. Because the y-axis of Figure 3 inherits the total number of <RR> intervals, this difference in slopes might be an artifact of the counting. Or not. The authors should mention the difference in slopes and give the readers at least a tentative explanation for it or a reason of why this is not relevant.
2. “Self-organized criticality explains the fractal patterns observed in heart rate variability and how a dynamic system operating at criticality can adapt to external changes more effectively than the principle of homeostasis”
To me, this is an overstatement. Firstly, the authors use the term homeostasis and cite references to this term that simply ignore a long-standing area of study known as allostasis and as rehostasis, i.e., the changes from the “static”. Secondly, the “homeostasis concept” along with its complementary allo/rehostasis ones and the physiological plasticity inherent to these concepts can, up to now, explain adjustments to diverse conditions faced during the life of an organism. Thirdly, the evidences of what is being “more effectively adjusted” than in the “homeostatic concept” are not given or even suggested. Fourthly, it is not clear to me the reasons why the authors oppose homeostasis and criticality: why does homeostasis exclude a system to operate near criticality?
3. “However, in two players, Zipf's law was clearly violated after the warm-up period. It could be hypothesized that these two players were overtrained. Identifying overtraining is a significant challenge in exercise physiology, but our study was not intended to address this question. “
This is a far-fetched hypothesis and none of the data support this statement. It should be deleted.
4. “However, the delay of four heartbeat intervals is remarkably close to the influence of breathing rhythm, which typically takes about four beats in most mammals, including humans [27].”
Reference 27, despite of its title, does not state this 4:1 entrainment between cardiac and respiratory frequencies for mammals. This reference discusses the possible role of sinus arrhythmia and of caridoventilatory coupling. Notice that while the former is directed related to heartrate variability, the latter is only a matter of matching frequencies. This matching is indeed somewhat fixed for mammals (and birds, as well) as the authors claim, but this is because both respiratory and cardiac frequencies scale similarly along size (body mass). Moreover, if this matching of frequencies (cardioventilatory coupling) is to be considered somehow related to the tipping point of the lines, then a deeper discussion should be presented on why the (first) line has a negative slope instead of a positive one since the 4:1 matching would head the bradycardic events.
Author Response
We appreciate the time and effort that the reviewer has put into evaluating our work.
Minor points:
- We no longer use adaptation; instead, we use adjustment.
- We now specify that the players are male subjects (line 67).
- We deleted the sentence “This number could be high as a single heartbeat interval may be involved in several bradycardias based on their duration” (lines 145-146).
- We deleted the sentences “Further studies are needed to better understand cardiovascular self-organized criticality and its pathophysiological implications. Fortunately, our study also shows that self-organized criticality can be studied easily with a simple sports watch that records heart rate, combined with a straightforward analysis. We identified, counted, and classified the spontaneous bradycardia on heart rate recordings” (lines 272-277).
Not so minor points:
- The slopes are now included in the text (lines 101; 165-174), Figure 4, and Figure 5, and along with a discussion about them (lines 265-270).
- We rewrote the sentence: Self-organized criticality explains the fractal patterns observed in heart rate variability and how a dynamic system operating at criticality can adjust to external changes more effectively than a system relying solely on regulatory loops with negative feedback [1,3-5,10] (Lines 219-222).
3 We deleted the sentence “It could be hypothesized that these two players were overtrained. Identifying overtraining is a significant challenge in exercise physiology, but our study was not intended to address this question“(Lines 283-285).
- We deleted the sentence and the linked paragraph of the discussion “However, the delay of four heartbeat intervals is remarkably close to the influence of breathing rhythm, which typically takes about four beats in most mammals, including humans [27]” (Line 289-295).
Reviewer 2 Report
In this study the Authors analyze heart rate recordings in professional soccer players in two positions (supine vs. standing), throughout a three-phase training period. The study design is compelling, considering the longitudinal nature of the recordings. Even though the manuscript presents some interesting findings, I think it can be improved on several levels. The analysis resorts to the bare minimum (only assessing changes in resting heart rate and verifying if a power-law relationship can be established between the length and frequency of bradycardia periods), without providing concise physiological implementation or substantial practical utility. It would be interesting to see how the presented results compare to those of a control group (i.e., individuals not participating in a training program). Furthermore, I think a more in-depth analysis of the currently available dataset could be carried out to further support the notion of self-organized criticality. I summarized my comments and suggestions below that I hope will help the Authors improve their manuscript.
- Even though power-law scaling is characteristic of systems near a critical state, power-law scaling in itself does not prove criticality. Can the Authors argue their hypothesis in more detail, i.e., why the presence of Zipf's law indicate a (self-organized) critical state, instead of merely the presence of long-term autocorrelation in heartbeat intervals? In the works of Bak, SOC is not only supported by the power-law relationship between avalanche size and frequency, but also waiting times between avalanches of similar size follow an exponential distribution. The Authors might also find interesting the works of CC Lo et al. (2002, 2004, 2013) investigating the plausible criticality of sleep-wake transitions. In those studies, not only a power-law relationship was established between the duration and frequency of brief awake periods, but also the waiting times between similar states (wake periods of equal length) followed an exponential (stochastic) distribution. A similar approach was employed by Racz et al. (2018, phys meas) to argue that connectivity dynamics in the prefrontal cortex reflect a critical state of brain function.
- Regarding the physiological interpretation of the results, the Authors might find interesting the work of Amaral et al. (2001) showing how sympathetic and parasympathetic blockade affect the scale-free/power-law characteristics of heart rate variability.
- Relatedly, even though the Authors confirm the presence of a power-law relationship, this relationship is not further characterized by e.g., the scaling exponent. It would be interesting to explicitly investigate if the scaling exponent (or in case of bimodal distributions as on Figure 3b, scaling exponents) change in response to the training program, or due to posture change. Also, it appears that on Figure 3c the second range the relationship rather follows an exponential and not a power-law function, which is not addressed in the manuscript. This might indicate a phase transition (plausibly induced by the 'warm up' phase) from a critical to non-critical state.
- Regarding the presence of a power-law relationship, the Authors might want to confirm it using more sophisticated methods such as that proposed by Clauset et al. (2009).
- Autonomic adaptation in this study is characterized on a longer, stationary setting (5-minute long recordings in supine and standing positions), however the fast adaptation period in the transitory state (going from supine to standing position) - that can also carry physiological relevance and reflect autonomic nervous system function - is not assessed here. In that regards, the Authors might find interesting the work of Mukli et al. (2021) investigating how the network of cardiorespiratory and cerebrovascular systems reorganize in response to orthostatic challenge.
- Can the Authors contrast their results with any behavioral output measures, to see if the power-law nature (or scaling exponent) of bradycardias is related to e.g., sports performance, or other physiological indices? This would greatly help the interpretability of results and could shed light in potential utility of computing such measures.
- It is not entirely clear to me, what the asterisk symbols indicate on Figure 2. In the statistical analysis it is stated that Friedman test was utilized, however Friedman test only indicates if there is a main effect related to the repeated measures factor. Are the individual asterisk symbols indicate that average heart rates at T2, T3 and T4 are significantly reduced compared to those at T0? Please clarify and indicate clearly, also specify the type of test used for post hoc pairwise comparisons. Also, the Friedman test (being a non-parametric statistical test) is generally utilized in case of non-normally distributed data, otherwise why not use repeated measures ANOVA? In case of normally distributed data (plus other assumptions of rmANOVA such as sphericity) the data is right to be reported as mean +- SEM, however in case of non-normal distributions (as indicated by the Friedman test) please report with median +- interquartile range, or similar.
The quality of English is sufficient, the Authors might want to double-check if they could improve phrasing in some instances (e.g., 40-41, two consecutive sentences start with 'This', which is unfortunate).
Author Response
We appreciate the time and effort that the reviewer has put into evaluating our work.
- The reviewer has suggested adding a control group that consists of individuals who are not involved in a training program. The proposed design by the reviewer is indeed elegant. However, it is not entirely clear whether the reviewer suggests a control group from the general population, or a control group composed of untrained professional players. We acknowledge that the average reader of Entropy may not be aware that professional soccer players are highly exceptional athletes who cannot be compared to the general population. We have now addressed this point in the manuscript (“Professional soccer players are high-performance athletes and cannot be compared to the general population” Lines 203-204). We hope the reviewer understands that it is not feasible to have a control group consisting of untrained professional soccer players due to the significant budgetary constraints associated with such a research program. Additionally, it is unlikely that any player would agree to jeopardize their career by taking a prolonged training break and sacrificing visibility.
- We were not previously aware that the waiting times between avalanches could serve as supporting evidence for self-organized criticality (SOC). We will investigate this aspect in cardiovascular time series. However, it should be noted that we would require significantly longer time series than those obtained in our experiment. We reference the studies by Lo et al. and Racz et al. in the new version of the manuscript and discuss this point. (“Lo et al. demonstrated self-organized criticality in sleep-wake rhythms by analyzing not only the distribution of brief awakening episodes but also the sleep duration between these episodes {29]. A similar approach was employed by Racz et al. to argue that connectivity dynamics in the prefrontal cortex reflect a critical state of brain function [21]. This kind of methods could potentially be applied to heart rate variability time series, but it would require significantly longer recordings compared to the 5-minute ones conducted in our study” Line 237-243).
- We have now included a mention of the reference by Amaral et al. (“Amaral et al. confirmed that heart rate variability complexity remains high in subjects who are motionless [32]” Lines 255-256).
- The slopes are now included in the text (lines 101; 165-174), Figure 4, and Figure 5, and along with a discussion about them (lines 265-270).
- We have now included a mention of the reference by Clauset et al. (“Conducting longer recordings in future studies would also provide an opportunity to vali-date the power law relationship using more advanced techniques, as proposed by Clauset et al., instead of relying solely on linear regression [30]” Lines 244-246).
- In a previous study, we investigated the transitional state between supine and standing to evaluate self-organized criticality. However, it is important to note that the laboratory setting used in that study is not feasible for a field study like the one presented in the current manuscript [24]. We have now included the reference by Mukli et al. [22] (“The analysis of transitions between two states, such as the shift from a supine to a standing position, can provide valuable insights, as suggested by Mukli et al. [22}. Unfortunately, the experimental setup required for such analyses was not feasible in the field experiment we are reporting here. However, we have previously demonstrated critical phase transitions during position shifts by replicating on the cardiovascular system the seminal study on movement control conducted by Shöner & Kelso [23, 24]”, Lines 227-233).
- We acknowledge the reviewer's point that the average reader of Entropy may not be familiar with the fact that performance cannot be objectively assessed in intermittent sports such as soccer. In such sports, performance relies more on abilities rather than aerobic capacity and is influenced not only by physical fitness but also by situational factors like tactics, scoring, and refereeing. To address this, we have included a reference [15] to support this point and provide further context (“Individual performance was not assessed in this experiment due to the absence of a known objective measurement to evaluate it within the context of a so-called team intermittent sport. In soccer, performance relies primarily on abilities rather than aerobic capacity and is influenced not only by physical fitness but also by situational factors such as tactics, scoring, and refereeing [15]”, Lines 85-89.
- We have completely revised the statistics. They are done with a linear mix model followed by tukey’s multiple comparison tests. Normality was assessed using d’Agostino-Pearson omnibus K2 tests. The statistic paragraph is totally rewritten to mention these points (“Data were presented as the mean ± standard error of the mean (SEM). Statistical analyses were conducted using Prism 8 software (GraphPad Software, San Diego, CA, USA). We considered that a Zipf's graph distribution fit a straight line when the absolute value of the correlation coefficient |r| exceeded 0.95, as this high cut-off is commonly used to test for power laws [7, 12, 13]. The normality of variable distributions was assessed using d’Agostino-Pearson omnibus K2 tests. Pearson correlations were performed when necessary. Linear mixed models were employed to model the relationship between position and training, followed by Tukey's multiple comparison tests when applicable. Statistical significance was set at p < 0.05”, Lines 104-116).
- We rewrote the sentence to avoid the double “this” (“Pile of sand was poised at equilibrium but near to criticality, which is referred to as self-organized criticality”, Line 40).
Round 2
Reviewer 2 Report
I would like to thank the Authors for addressing most of my points. I understand that some of the suggestions could not be easily resolved, or would have required substantial work that might be beyond the scope of the current study (e.g., inclusion of a control group). I still find the manuscript slightly lacking, however this does not discredit the results that are exhibited here. Also, some of my comments were rather suggestions for future work instead of critique of what is currently presented. Nevertheless, I believe this set of findings is worth publishing as is.
I have no further comments.
Author Response
no